# Single Nucleotide Polymorphisms of Toll-like Receptor 4 in Hepatocellular Carcinoma—A Single-Center Study

**DOI:** 10.3390/ijms23169430

**Published:** 2022-08-21

**Authors:** Theodoros Androutsakos, Athanasios-Dimitrios Bakasis, Abraham Pouliakis, Maria Gazouli, Christos Vallilas, Gregorios Hatzis

**Affiliations:** 1Department of Pathophysiology, Medical School, National and Kapodistrian University of Athens, 11527 Athens, Greece; 22nd Department of Pathology, University General Hospital Attikon, Medical School, National and Kapodistrian University of Athens, 12462 Athens, Greece; 3Department of Basic Medical Sciences, Laboratory of Biology, Medical School, National and Kapodistrian University of Athens, 11527 Athens, Greece

**Keywords:** Toll-like receptor 4, hepatocellular carcinoma, liver cirrhosis, non-alcoholic steatohepatitis, alcoholic liver disease

## Abstract

Hepatocellular carcinoma (HCC) is the most common primary liver tumor leading to significant morbidity and mortality; its exact genetic background is largely unrecognized. Toll-like receptor-4 (TLR4) reacts with lipopolysaccharides, molecules found in the outer membrane of Gram-negative bacteria. In damaged liver, TLR4 expression is upregulated, leading to hepatic inflammation and injury. We tried to investigate the role of the two most common single-nucleotide polymorphisms (SNPs) of TLR4 in HCC-genesis. Aged > 18 years old, cirrhotic patients were included in this study. Exclusion criteria were non-HCC tumors and HIV co-infection. TLR4 SNPs association with HCC occurrence was the primary endpoint, and associations with all-cause and liver-related mortality, as well as time durations between diagnosis of cirrhosis and HCC development or death and diagnosis of HCC and death were secondary endpoints. A total of 52 out of 260 included patients had or developed HCC. TLR4 SNPs showed no correlation with primary or secondary endpoints, except for the shorter duration between HCC development and death in patients with TLR4 mutations. Overall, TLR4 SNPs showed no correlation with carcinogenesis or deaths in patients with liver cirrhosis; patients with TLR4 SNPs that developed HCC had lower survival rates, a finding that should be further evaluated.

## 1. Introduction

Liver cancer is the sixth most common cancer and the sixth cause of cancer-related death worldwide [1]. Hepatocellular carcinoma (HCC) is the most frequent primary liver cancer, accounting for more than 80% of all cases [2]. Major risk factors for HCC include cirrhosis (with 80–90% of HCCs developing in cirrhotic patients), chronic hepatitis B virus (HBV) infection (even without significant hepatic fibrosis), co-infection of hepatitis viruses with human immunodeficiency virus (HIV), liver damage due to aflatoxin, non-alcoholic steatohepatitis (NASH), family history and genetic factors [3,4]. Liver diseases are characterized by tissue inflammation with subsequent hepatocellular death, leading to continuous immune activation. Even though immune activation contributes to the restoration of tissue function, a heightened or prolonged immune response may lead to the replacement of hepatic parenchyma by fibrotic tissue and vascular architectural distortion, leading to liver cirrhosis and probably HCC [5,6]. It is now known that pro-inflammatory factors, e.g., transforming growth factor beta (TGF-β), interleukin-6 (IL-6) and tumor-necrosis factor alpha (TNF-a) play an important role in carcinogenesis [7,8,9]. This procedure, first postulated by Virchow back in 1863, is now called the “hepatic inflammation–fibrosis–cancer axis” [7].

Toll-like receptors (TLRs) are a family of pattern-recognition receptors that play a critical role in the activation of the innate immune system by recognizing pathogen-associated molecular patterns (PAMPs). To date, thirteen mammalian TLRs have been identified; from those, TLRs 1-11 are expressed in humans [10]. TLR-4 is one of the most studied TLRs and plays a key role in innate immunity by provoking inflammatory responses to its main ligand, lipopolysaccharides (LPS); molecules found in the outer membrane of Gram-negative bacteria [10]. In liver parenchyma, TLRs are expressed on Kupffer cells, dendritic cells, hepatic stellate cells, endothelial cells, and hepatocytes [11]. In healthy liver, TLR4 is expressed at a relatively low level, but when the liver is damaged, TLR4 expression is upregulated, and TLR4 signaling seems to perpetuate hepatic inflammation and tissue injury [12,13].

A variety of mice and cell line-based studies have associated TLR4 overexpression with HCC development [14,15,16,17,18,19,20,21,22], while, in humans, most studies have shown increased expression of TLR4 in HCCs, especially in more aggressive types, even though this finding is not ubiquitous [13,23,24,25,26,27].

The ability of TLRs to respond properly to their ligands may be impaired by single-nucleotide polymorphisms (SNPs) within TLR genes. In Caucasians, the two most commonly found SNPs are a transition at SNP rs4986790 that causes an Asp/Gly polymorphism at amino acid 299 and a transition at SNP rs4986791 that causes a Thr/Ile polymorphism at amino acid 399. Both SNPs can be found in up to 10% of the European population, showing a west–east gradient of the haplotype prevalence [27].

We have conducted a prospective observational study seeking to elucidate the role of Asp299Gly and Thr399Ile TLR4 SNPs in HCC in a cohort of Greek cirrhotic patients.

## 2. Results

### 2.1. Patients’ Characteristics

During the study period, a total of 262 patients were assessed for inclusion in the study. Of them, 260 patients were included, with 175 (67.3%) being males. Only two patients were excluded, one due to concomitant HIV infection and one due to patient’s unwillingness to take part in the study. The median age of participants was 65 years old (range: 25–88). Alcohol abuse was the most common cause of cirrhosis (83 patients, 31.9%), followed by HCV infection (49 patients, 18.9%), HBV infection (40 patients, 15.4%) and NASH (34 patients, 13.1%). All diagnoses leading to cirrhosis in our cohort are presented in Table 1.

In terms of laboratory testing, median alanine aminotransferase (ALT), aspartate aminotransferase (AST), gamma glutamyl-transferase (gGT) and alkaline phosphatase (ALP) serum levels were 32, 47, 71 and 116 IU/mL, respectively. The median international normalized ratio (INR) was 1.34 and median total serum bilirubin value was 1.35 mg/dL (Table 1). Median model for end-stage liver disease (MELD) score was 12 and median Child–Turcotte–Pugh (CTP) score was 7, with 105 patients having a CTP A staging, 96 CTP B and 59 CTP C, at study entrance.

Forty-one patients (15.8%) were diagnosed with HCC upon admission, and 11 additional (4.2%) developed HCC during their follow up. During the study period, 116 (44.6%) patients passed away. Seventy-four deaths (63.8% of all deaths) were deemed to be due to liver-related complications, while 35 (30.2% of all deaths) were attributed to other causes, such as non-HCC tumors, pneumonia, urinary tract infection and cardiovascular-related issues. The cause of death was unknown for seven patients.

The median follow-up time from study inclusion date was 13 months (range 0–125), when including 44 (16.9%) patients that were lost to follow up or died shortly after study inclusion. A total of 143 (55%) patients were monitored for at least 1 year, 109 (41.9%) reached 2 years, 89 (34.2%) patients 3 years and 53 (20.4%) patients 5 or more years.

The median time from diagnosis of cirrhosis until the end of follow up, HCC diagnosis and death were 58 (range 0–250), 17.5 (range 0–204) and 50 months (range 1–250), respectively. In 56 patients (21.5%), the diagnosis of cirrhosis was set upon study entry. From the 52 patients diagnosed with HCC, 44 died during the follow-up period with a median time, from HCC diagnosis to death, of 8 months (range 1–60).

### 2.2. TLR4 Genotyping

TLR4 genotyping revealed that 62 (23.85%) patients had a SNP in Asp299Gly (49 (18.85%) patients heterozygous and 13 (5%) homozygous), while 8 (3.08%) had an SNP in Thr399Ile, all heterozygous. Two patients had SNPs in both TLR4 299 and 399 (Table 1).

### 2.3. TLR4 SNPs and HCC Occurrence

From the 52 patients with HCC, 10 (19.2%) had TLR4 mutations (7 Asp299Gly, 2 Thr399Ile and 1 both Asp299Gly and Thr399Ile), while from the 208 patients with no HCC, TLR4 mutation was found in 62 (29.8%) (55 Asp/299Gly, 6 Thr399Ile and 1 both Asp299Gly and Thr399Ile) (*p* value 0.127).

In multivariable analysis, male gender and age showed a positive correlation with HCC development; on the contrary, neither TLR4 Asp299Gly nor Thr399Ile SNPs showed any correlations with HCC occurrence, either analyzed separately or together (Table 2). Similarly, cause of cirrhosis was not associated with HCC in univariable or multivariable analysis.

### 2.4. TLR4 SNPs and Secondary Endpoints

#### 2.4.1. All-Cause and Liver-Related Deaths

Out of the 116 patients that died during follow up, 33 had a mutation in TLR4 (25 Asp299Gly, 6 Gly299Gly, 1 Thr399Ile and 1 both Asp299Gly and Thr399Ile). Regarding the secondary end points, the multivariable analysis including gender, age, CTP and MELD scores, TLR4 mutations and cause of cirrhosis, all-cause death showed a statistically significant correlation with CTP (odds ratio (OR) 1.50 (95% confidence interval (CI) 1.21–1.86), *p* value 0.0002), while liver-related deaths were also correlated with CTP (OR 1.33 (95% CI 1.1–1.65), *p* value 0.0089). Both TLR mutations showed no association with either liver-related or any-cause deaths.

#### 2.4.2. Causes of Death in HCC Patients and Duration between HCC Diagnosis and All-Cause Death

A total of 46 patients with HCC died during the follow up, with eight of them having the aforementioned TLR4 SNPs. Regarding patients with TLR4 SNPs, the most common causes of death were sepsis, hepatic coma and variceal bleeding (each with two patients); in patients with wild-type TLR4, hepatic coma with 16 patients and sepsis with six patients were the most common causes of death; no statistically significant differences were found between patients with and without TLR4 SNPs (Appendix A). Examining survival in patients with HCC with regard to TLR4 mutations, the median time between HCC diagnosis and all-cause death for those with TLR4 mutations was 4 months (range 1–6) and for those without, 9 months (range 1–60). The relevant survival curves are depicted in Figure 1 showing that survival time for patients with TLR4 mutations is significantly lower (Log rank test *p* = 0.0058). When comparing patients developing HCC with and without TLR4 SNPs in terms of age, liver function tests, MELD and CTP scores, no differences were found (Appendix A). In a multivariable model involving TLR4 mutations, age, gender, MELD and CTP scores and cause of cirrhosis, TLR4 mutations showed a statistically significant correlation with overall survival (hazard ratio (HR): 0.327, *p* value 0.0271), and similarly MELD (HR: 0.911, *p* value 0.0264) and CTP (HR: 1.422, *p* value 0.0200); on the contrary, no other factor proved to have a role in HCC patient survival.

#### 2.4.3. Duration between HCC Diagnosis and Liver Related Death

In relation to the survival of patients with HCC that died from liver-related causes, the median time between HCC diagnosis to liver-related death was 8 months (Q1–Q3: 6–13), for those with TLR4 mutations was 4 months (Q1–Q3: 1–6) and for those without, 12 months (Q1–Q3: 7–15). The relevant survival curves are depicted in Figure 2 showing that survival time for patients with HCC TLR4 mutations is significantly lower (Log rank test *p* = 0.0021). Furthermore, we constructed a multivariable model involving TLR4 mutations, age, gender, MELD CTP scores and cause of cirrhosis. TLR4 mutations showed a statistically significant correlation with overall survival (HR: 0.333, *p* value 0.0335); in contrast, age, gender, MELD CTP scores and cause of cirrhosis were not found to have a significant role.

#### 2.4.4. Duration between Diagnosis of Cirrhosis and HCC Development or Death

Regarding the time duration between diagnosis of cirrhosis and HCC, no correlation to TLR4 SNPs was found (*p* = 0.7943). More precisely, the median time between cirrhosis and HCC diagnoses was 47 months (range 11–160) for patients with TLR4 mutations and 46 (range 1–204) for those without. In a multivariable approach involving TLR4 mutations, MELD, CTP, patients’ age, gender and cause of cirrhosis, gender proved to be a statistically significant factor (*p* = 0.0123) with a HR for females to males of 0.28, as well as MELD (HR: 0.92, *p* = 0.0313) and CTP scores (HR: 1.40, *p* = 0.0140).

Similarly, concerning duration between diagnosis of cirrhosis and all cause death, no significant correlation to TLR4 SNPs was found for the patients with TLR4 mutations (median: 48, range: 3–166) and for those without (median: 40, range: 1–210) (*p* = 0.8001). The multivariable approach, involving TLR4 mutations, MELD, CTP, presence of HCC, patients’ age, gender and cause of cirrhosis showed that gender was the sole statistically significant factor (*p* = 0.0251), with a HR for females to males of 0.55. Table 3 presents detailed results of the univariate and multivariate analysis for the role of the previous parameters in the time from cirrhosis development until death.

## 3. Discussion

TLR-4 plays an important role in the proper function of the human innate immune response, mainly by recognition of PAMPs. The ability of TLR-4 to respond properly to their ligands may be impaired by various SNPs. TLR4 Asp299Gly and Thr399Ile are the two most commonly found TLR4 SNPs in Caucasians. These SNPs have been associated with many diseases, including a variety of cancers, infections, inflammatory bowel disease, chronic obstructive pulmonary disease, asthma, and diabetes mellitus, showing either protective or aggravating function; however, most studies are relatively small, and results are contradictory [28,29,30,31,32,33,34,35]. Likewise, the aforementioned TLRs have been studied in a variety of liver diseases, such as hepatitis C, cirrhosis and HCC, with no definite role being established [25,36,37].

To study the significance of TLR4 activation in HCC, we examined the two most common TLR4 SNPs in a cohort of 260 cirrhotic patients and followed them up for up to 125 months for possible HCC development. A significant proportion of our patients (68 patients, 27.2%) had at least one SNP among the two examined loci, and in the vast majority of them, an Asp299Gly mutation was found (62 patients). The exact prevalence of both SNPs in the Greek population is largely unknown since only a handful of studies regarding that issue exist. In these studies, the prevalence of Asp299Gly and Thr399Ile SNPs range, in various populations, from 5–25% and from 2–13%, respectively [38,39,40,41].

A total of 52 patients (20%) had either HCC on inclusion or developed HCC during follow up. Age and male gender were strongly correlated with HCC incidence, while none of the studied SNPs showed any association with HCC development. This finding is in accordance with previously published studies showing that both age and male gender are aggravating factors for the development of HCC [42,43]. Likewise, TLR4 mutations showed no statistically significant correlation with duration of cirrhosis or HCC development. Regarding TLR4 SNPs and the occurrence of HCC, study results are contradictory, with various SNPs being associated with either increased or decreased probability for HCC development [36,37,44,45,46,47,48,49]. Likewise, the results are not consistent when Asp299Gly and Thr399Ile are associated with HCC development [36,44,45,46]. The lack of definite correlation between TLR4 SNPs and HCC development supports the diversity of liver carcinogenesis, where in lack of TLR4-based inflammation, other pathways may be activated.

TLR4 SNPs failed to show any correlation with all-cause and liver-related death incidence. The vast majority of our patients died due to liver cirrhosis-related complications, with variceal bleeding and spontaneous bacterial peritonitis (SBP) being the two most frequent causes. Since these complications are found most commonly in patients with deteriorating liver function, it is not peculiar that CTP scoring was the most important factor correlated with liver-related death. Limited data exist regarding TLR4 and SBP. In two studies, one including both D299G and T399I and the other only the D299G SNP, TLR4 SNPs were not found to be risk factors for SBP development [50,51]. Given the fact that SBP was a major cause of liver-related death in our cohort, the lack of association between TLR4 SNPs and liver-related mortality is not surprising. As far as non-liver-related mortality is concerned, sepsis, either due to pneumonia or urinary tract infection, was the most common cause of death. Even though older studies have associated the presence of, mainly, D299G with increased incidence of Gram (−) infections and worse prognosis of sepsis, these findings were not validated in later studies [52,53,54,55,56]. Likewise, in our cohort, TLR4 SNPs were not correlated with all cause death. However, CTP scoring was related with all-cause deaths, highlighting the poor performance status of patients with advanced liver cirrhosis.

Patients with TLR4 mutations had a more aggressive course of HCC, with a significantly shorter time interval between HCC diagnosis and death. Given the fact that the two studied mutations lead to a diminished response to inflammation, this finding at a first glance seems illogical, since multiple studies in mouse models and cell lines have shown that constant inflammation, induced mainly by lipopolysaccharides (LPS), leads to tumorigenesis [15,21,57]. On the contrary, Wang Z et al. showed that, in mouse models, TLR4 activity protected against HCC progression by regulating expression of DNA repair protein Ku70, a protein encoded by the X-ray repair cross complementing 6 (XRCC6) gene [16,58]. In another study from Hsu CM et al, in 298 patients with HCC, lower XRCC6 mRNA and protein expressions were found in HCC tissues compared with non-HCC ones, implicating a protective role of TLR4 in HCC development [59]. One possible scenario for our findings would be that upon HCC initiation, the decrease in TLR4-related DNA repair proteins, such as Ku70, and the diminished activity of the so-called senescence response to defense against tumorigenesis in the liver could lead to stimulation of cancerous cell proliferation, attenuation of autophagy and programmed cell death, and promotion of malignant transformation, thus leading to a more aggressive HCC phenotype [16,58,60].

The main limitation of our study is the low number of individuals with HCC, since only 52 patients had or developed HCC; thus, our findings should be further evaluated in larger studies. Moreover, our study lacks evaluation of histopathological findings between patients with and without TLR4 SNPs; this is due to the fact that HCC diagnosis can be established with typical radiological findings in patients with cirrhosis; thus, only a handful of patients underwent liver biopsy. Another limitation is the fact that our patients have a variety of causes leading to cirrhosis; however, since cirrhosis per se is a major predisposing factor for HCC development, and we have performed sub-analysis for these causes, we feel this is not a major limitation.

## 4. Materials and Methods

### 4.1. Inclusion Criteria and Definitions

All patients visiting the Hepatology outpatient clinic of Pathophysiology department of “Laikon” University Hospital during a 6-year period, from December 2010 to December 2016, were eligible for recruitment in the study. Inclusion criteria included the presence of liver cirrhosis, confirmed by liver biopsy and/or transient elastography, and patient age over 18 years. Exclusion criteria included the presence of non-HCC malignancies or HIV co-infection. The primary endpoint of the study was the association of TLR4 SNPs with HCC occurrence, while secondary endpoints were association of TLR4 SNPs with all-cause and liver-related mortality and time durations between diagnosis of cirrhosis and HCC development, diagnosis of cirrhosis and death and diagnosis of HCC diagnosis and death.

All patients included in this study were followed up every 6 months until study completion in December 2020.

HCC diagnosis was made according to EASL guidelines [4] either via liver biopsy or typical radiographic findings in computed tomography or magnetic resonance imaging. Tumors not fulfilling these criteria were regarded as non-HCC. Liver-related death was considered any death occurring from complications of liver cirrhosis, such as SBP, variceal bleeding, hepatorenal syndrome or hepatic coma. MELD and CTP calculations and grading were performed according to the literature [61,62].

### 4.2. TLR Genotyping 

TLR genotyping was performed in peripheral blood. More specifically, genomic DNA from peripheral blood was isolated using a NucleoSpin Blood kit (Macherey–Nagel, GmbH & Co. KG, Düren, Germany) according to the manufacturer’s instructions. Genotyping in all selected polymorphisms was performed by PCR-RFLP analysis for the TLR-4 Asp299Gly (rs4986790) and TLR-4 Thr399Ile (rs4986791) polymorphisms, as previously described [63]. Specific, primers for TLR-4 Asp299Gly were: forward (5′-GATTAGCATACTTAGACTACTACCTCCATG-3′) and reverse (5′-GATCAACTTCTGAAAAAGCATTCCCAC-3′) and for TLR-4 Thr399Ile were forward (5′-GGTTGCTGTTCTCAAAGTGATTTTGGGAGAA-3′) and reverse (5′-CCTGAAGA CTGGAGAGTGAGTTAAATGCT-3′). The underlined bases in both forward primers indicate an altered nucleotide that was introduced in order to create either a NcoI (TLR-4 Asp299Gly) or a HinfI (TLR-4 Thr399Ile) (both from New England BioLabs, Ipswich, MA, USA) restriction site, respectively. PCR reactions were run at 95 °C for 5 min followed by 35 cycles at 95 °C for 30 s, 55 °C for 30 s, 72 °C for 30 s, and a final incubation at 72 °C for 5 min. A 15 μL aliquot of the product was digested with the appropriate restriction enzyme and electrophoresed in a 3% agarose gel to identify the TLR-4 alleles on the basis of the respective allele size. After digestion, fragment sizes for carriers of the polymorphic allele decreased from 249 bp (wild-type) to 223 bp for the 299 residue and from 406 bp (wild-type) to 377 bp for the 399 residue.

### 4.3. Statistical Analysis

Statistical analysis was performed via the SAS for Windows 9.4 software platform (SAS Institute Inc., Cary, NC, USA). Descriptive values were expressed as median and range and for the categorical data as frequencies and the relevant percentages. Comparisons between groups for the qualitative parameters were made using the chi-square test (and if required, Fisher exact test was performed, i.e., when the expected frequency in the contingency table was <5 in ≥25% of the relevant cells). We applied multivariable analysis using the backward elimination logistic regression method. For statistical reasons, we grouped cirrhosis etiology into 5 categories: (a) NASH-induced; (b) alcoholic; (c) viral, including HBV and HCV infections; (d) autoimmune, including autoimmune hepatitis (AIH), primary biliary cholangitis (PBC) and AIH/PBC overlap syndromes; and (e) other, including the remaining etiologies. For the evaluation of survival time of patients with HCC in relation to TLR mutations, we evaluated the survival curves using the Log-rank method, and for multivariate analysis of survival time, we applied regression analysis of survival data based on the Cox proportional semi-parametric hazards model (procedure PHREG). The effect of explanatory variables was evaluated using the hazard rates and the relevant *p* values. The significance level (*p* value) was set to <0.05, and all tests were two sided.

## 5. Conclusions

Overall, in our study, TLR4 SNPs showed no correlation with carcinogenesis or deaths in patients with liver cirrhosis, further supporting the inability of TLR4 SNPs to act as biomarkers for HCC. However, patients with TLR4 SNPs that developed HCC had lower survival rates, a finding that should be further evaluated.

## Figures and Tables

**Figure 1 ijms-23-09430-f001:**
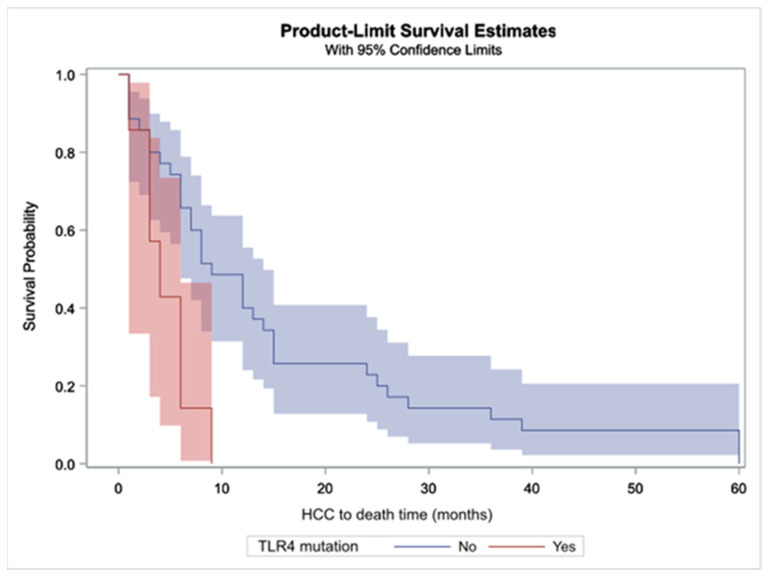
Survival curves for the patients with HCC in relation to TLR4 mutations. Shaded area indicates the 95% confidence interval.

**Figure 2 ijms-23-09430-f002:**
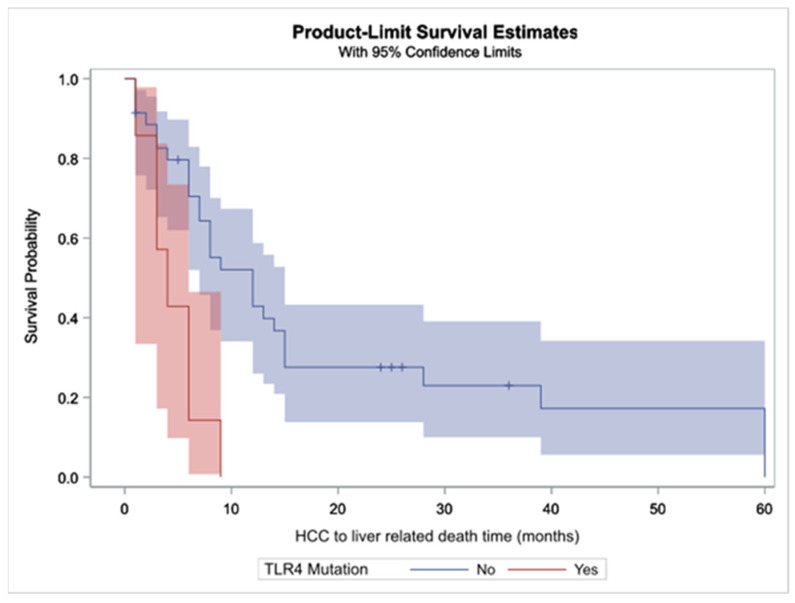
Survival curves for liver-related deaths for the patients with HCC in relation to TLR4 mutations. Shaded area indicates the 95% confidence interval.

**Table 1 ijms-23-09430-t001:** Baseline characteristics of the study population.

Patient Characteristic	Value
Age in years, median (range)	65 (25–78)
Gender male, n (%)	175 (67.3)
AST (IU/mL), median (range)	47 (29–75)
ALT (IU/mL), median (range)	32 (20–52)
gGT (IU/mL), median (range)	71 (38–156)
ALP (IU/mL), median (range)	116 (83–179)
Total bilirubin (mg/dL), median (range)	1.35 (0.86–2.94)
INR median (range)	1.34 (1.2–1.6)
MELD score, median (range)	12 (9–18)
CTP score, median (range)	7 (6–9)
CTP staging (A/B/C), n (%)	105/96/59, (40.4/37/22.6)
HBV-induced cirrhosis, n (%)	40 (15.4)
HCV-induced cirrhosis, n (%)	49 (18.9)
NASH-induced cirrhosis, n (%)	34 (13.1)
ASH-induced cirrhosis, n (%)	83 (31.9)
PBC-induced cirrhosis, n (%)	17 (6.5)
AIH-induced cirrhosis, n (%)	8 (3.1)
PBC/AIH-induced cirrhosis, n (%)	7 (2.7)
Cryptogenic cirrhosis, n (%)	10 (3.8)
TLR4 299 heterozygous/homozygous SNP, n (%)	49/13, (18.9/5.0)
TLR4 399 SNP heterozygous/homozygous SNP, n (%)	8/0 (3.1/0)

**Table 2 ijms-23-09430-t002:** Uni- and multivariable analyses for HCC development. *p* values in bold indicate statistical significance.

	Univariable Analysis		Multivariable Analysis	
Variable	OR (95% CI)	*p* Value	OR (95% CI)	*p* Value
Age	1.03 (1–1.05)	0.0692	1.03 (1–1.07)	0.0348
Gender (Male)	2.4 (1.12–4.99)	0.0234	2.7 (1.07–6.83)	0.0358
NASH	1.64 (0.38–7.11)	0.3204	1.44 (0.31–6.78)	0.5915
Alcohol	1.08 (0.28–4.22)	0.8873	0.9 (0.21–3.87)	0.468
Viral	1.88 (0.5–7.03)	0.0653	1.88 (0.46–7.66)	0.1137
Autoimmune	0.55 (0.1–3.06)	0.1624	0.88 (0.13–5.89)	0.6357
MELD	1.01 (0.98–1.05)	0.4620	1 (0.93–1.07)	0.8892
CTP	1.07 (0.95–1.21)	0.2761	1.13 (0.89–1.43)	0.3137
TLR4 Mutations (both)	0.6 (0.3–1.31)	0.2071	0.08 (0–3.2)	0.1812
TLR4 299 Mutation (any)	0.5 (0.23–1.17)	0.1141	5.29 (0.15–183.4)	0.3575
TLR4 399 Mutation (any)	2.4 (0.6–10.7)	0.2231	17.7 (0.76–414.14)	0.0741

**Table 3 ijms-23-09430-t003:** Uni- and multivariable analyses for the time from cirrhosis until death. *p* values in bold indicate statistical significance.

	Univariate Analysis		Multivariate Analysis	
Parameter	Hazard Ratio	*p* Value	Hazard Ratio	*p* Value
TLR4 mutation	0.802	0.2690	0.717	0.1374
MELD	1.008	0.3694	0.993	0.6746
CTP	1.024	0.5024	1.078	0.2582
HCC	1.382	0.0936	1.314	0.1832
Age	1.007	0.3044	1.004	0.5434
Gender (female)	0.647	0.0276	0.555	**0.0251**
Cause of cirrhosis				
NASH	0.768	0.2275	0.645	0.4233
Alcohol	0.905	0.6224	0.437	0.1067
Viral	1.287	0.1851	0.441	0.1125
Autoimmune	1.096	0.7273	0.712	0.5662

## Data Availability

Data used to support the findings of this study are found in the article. If more data are needed, they are available from the corresponding author upon reasonable request.

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
