# Peer review of "Single Nucleotide Polymorphisms of Toll-like Receptor 4 in Hepatocellular Carcinoma—A Single-Center Study"

_ijms, 2022, doi:10.3390/ijms23169430_

Round 1
Reviewer 1 Report
There are some minor comments.
It would be better to explain the materials and method in detail.
- Was the TLR4 genotyping test performed on tumor tissue or
non-neoplastic tissue?
- How were non-HCC cases diagnosed (for example, on the basis of
histopathology)?
It would be better to describe whether there are any differences in
pathological findings between HCC with TLR4 mutations and HCC without
TLR4 mutations.
Please check English grammar.
most studies have showed -> most studies have shown
Wang Z et al, -> Wang Z et al.
Author Response
Thank you very much for your comments. Please find below the authors' reply in each one.
- It would be better to explain the materials and method in detail.
- Was the TLR4 genotyping test performed on tumor tissue or non-neoplastic tissue?
Response:
Thank you very much for your comment. TLR genotyping was performed in peripheral blood only. To clarify that a new sentence is added in “Materials and methods” section at subsection 4.2
- How were non-HCC cases diagnosed (for example, on the basis of histopathology)?
Response:
Thank you for your comment. Since HCC is the only tumor that can be diagnosed without liver biopsy, all non-HCC tumors were diagnosed through biopsy. To better clarify that we have added in materials and methods section the sentence “Tumors not fulfilling these criteria were regarded as non-HCC” right after explaining how HCC diagnosis was made, at subsection 4.1.
- It would be better to describe whether there are any differences in pathological findings between HCC with TLR4 mutations and HCC withoutTLR4 mutations.
Response:
Thank you for your comment. Since liver biopsy is not mandatory for HCC, we had only a handful HCC biopsies, so no analysis was performed. We have added this also as a limitation in discussion section.
- Please check English grammar.
most studies have showed -> most studies have shown
Wang Z et al, -> Wang Z et al.
Response:
Thank you for your comment. The manuscript has been re-evaluated from all authors for grammatical mistakes.
Reviewer 2 Report
This is a clinical study in which the authors have investigate the role of the two most common single-nucleotide polymorphisms of TLR4 in HCC. However, the number of patients evaluated appears to be low. The number of patients should be increased to obtain clinically relevant data. In addition, it is not clear that why TLR4 SNPs showed no direct correlation with primary or secondary endpoints. Moreover, why patients with TLR4 SNPs that developed HCC had lower survival rates has not been analyzed. Overall, the study is too preliminary in nature and not suitable for publication in IJMS.
Author Response
Thank you very much for your remarks. Please find below the authors' reply.
1. This is a clinical study in which the authors have investigate the role of the two most common single-nucleotide polymorphisms of TLR4 in HCC. However, the number of patients evaluated appears to be low
Response:
Thank you for your comment. As already mentioned as limitation in discussion section the number of patients developing HCC in our cohort was low (52 patients) so some of our results could be influenced by that.
2. The number of patients should be increased to obtain clinically relevant data. In addition, it is not clear that why TLR4 SNPs showed no direct correlation with primary or secondary endpoints
Response:
As far as the primary endpoint of our study is concerned, only a few studies have addressed this question in the literature with non-consistent results. We have added this in the discussion section with the appropriate references.
Regarding the correlation between TLR4 and secondary endpoints, our results are in line with previous studies. More specifically, most of our patients died due to sepsis or spontaneous bacterial peritonitis (SBP). While for SBP TLR4 SNPs don’t seem to play a role, the results regarding sepsis are in large contradictory. This is now also added in discussion section with the appropriate references.
3. . Moreover, why patients with TLR4 SNPs that developed HCC had lower survival rates has not been analyzed
Response:
To further analyze the survival rates of patients with HCC and TLR4 SNPs we have added an appendix in our manuscript, with 2 additional tables. In the first table causes of death of patients developing HCC can be found. In this table one can see that no statistical differences regarding the cause of death between patients with TLR4 SNPs and those without are found. In the second table baseline characteristics of the 2 above-mentioned groups are found, again with no differences. We have also added these data in the results section, paragraph 2.4.2
Lastly, we have added a possible explanation of our findings regarding the survival rates in patients with TLR4 SNPs in the discussion section, highlighting that a possible reason could be the decrease of TLR4-dependent DNA-repair proteins.
Round 2
Reviewer 2 Report
The authors have addressed all my concerns.